# Towards a Smart Smoking Cessation App: A 1D-CNN Model Predicting Smoking Events

**DOI:** 10.3390/s20041099

**Published:** 2020-02-17

**Authors:** Maryam Abo-Tabik, Nicholas Costen, John Darby, Yael Benn

**Affiliations:** 1Department of Computing and Mathematics, Faculty of Science and Engineering, Manchester Metropolitan University, Manchester M15 6BH, UK; n.costen@mmu.ac.uk (N.C.); J.Darby@mmu.ac.uk (J.D.); 2Department of Psychology, Manchester Metropolitan University, Manchester M15 6GX, UK; Y.Benn@mmu.ac.uk

**Keywords:** smoking cessation app, smoker’s behaviour, addictive behaviour, machine learning, deep learning, CNN, control theory

## Abstract

Nicotine consumption is considered a major health problem, where many of those who wish to quit smoking relapse. The problem is that overtime smoking as behaviour is changing into a habit, in which it is connected to internal (e.g., nicotine level, craving) and external (action, time, location) triggers. Smoking cessation apps have proved their efficiency to support smoking who wish to quit smoking. However, still, these applications suffer from several drawbacks, where they are highly relying on the user to initiate the intervention by submitting the factor the causes the urge to smoke. This research describes the creation of a combined Control Theory and deep learning model that can learn the smoker’s daily routine and predict smoking events. The model’s structure combines a Control Theory model of smoking with a 1D-CNN classifier to adapt to individual differences between smokers and predict smoking events based on motion and geolocation values collected using a mobile device. Data were collected from 5 participants in the UK, and analysed and tested on 3 different machine learning model (SVM, Decision tree, and 1D-CNN), 1D-CNN has proved it’s efficiency over the three methods with average overall accuracy 86.6%. The average MSE of forecasting the nicotine level was (0.04) in the weekdays, and (0.03) in the weekends. The model has proved its ability to predict the smoking event accurately when the participant is well engaged with the app.

## 1. Introduction

Smoking is considered one of the leading causes of deaths internationally. According to a recent NHS report [1], smoking caused the deaths of approximately 7900 people in England alone in 2016. The report further states that smoking is not only harmful to the smokers, but many diseases may be caused by the exposure to passive smoking, especially affecting children who are particularly vulnerable to the effects of passive smoking. This makes reducing cigarette smoking a significant public health priority. To support efficient and timely delivery of intervention for those wishing to quit smoking, it is important to be able to model the smoker’s behaviour, and in order to do that, it needs to target both endogenous stressors (e.g., nicotine effect, craving, etc.)and exogenous stressors (e.g., timing, location, type of activity, etc.) that trigger the smoking events [2].

With advances in technology, new possibilities have emerged for creating efficient cessation programs, particularly through the use of mobile apps. This new technology has many advantages over traditional therapies; it can reach people wherever they are; enhance their experience by opening new channels between the therapist and the smoker; lastly, it offers the possibility to access databases that can provide individual feedbacks on the smokers’ current status [3], Several methods have been used to provide intervention using mobile apps, For example, text messages either in regular or randomized intervals, or by making the user initiate access to the intervention by reporting on indicators that may cause a potential lapse [4,5,6].

Investigations using self-reporting as a method have indicated that the reported predictors can provide a high degree of possibility for predicting potential lapses [4,5]. Schick et al. [6] improved this method by using Hidden Markov Models to set patterns for the timing and places in which individuals are most likely to smoke, and then use these patterns for better delivery of the support messages. This paper did not report any analytical results that are related to Hidden Markov Models, but rather focused on the positive feedback from the participants who used their mobile application.

Recent advances in computation make machine learning a perfect tool for modelling smokers’ behaviour, enabling the implementation of smart mobile apps that have the ability to provide ‘just in time’ intervention. For example Dumortier et al. [7] used machine learning methods to evaluate the urge to smoke based on participant reporting of 41 features (e.g., alcohol consumption, mood status, hunger, location, type of working, etc.) that may trigger an urge to smoke. They compared three different machine learning algorithms (naive Bayes classifier, discriminant analysis classifier, and decision tree learning), and checked the accuracy of the classification based on a number of selected features. Results indicated that machine learning had the ability to estimate the smokers’ urge rating with an accuracy of the classifications up to 86%. However, the models relied on the users reporting a large number of input features. Another study [8] also used decision tree to predict daily smoking behaviour. Here population information from the 2015 China Adult Tobacco Survey Report was used; the research modelled an equation that calculates the probability of smoking time based on gender, age and time and used statistical information from the dataset as well as some additional extracted features as input to the decision tree model. The researchers concluded that the best method of prediction is XGBoost with 84.11% accuracy.

In addition to the issues around self reporting, most existing apps for smoking cessation do not take into consideration the complexity of nicotine dependence treatment or the specific needs of the users [3]. Self-reporting as a method can be inaccurate as it is sensitive to self-biased errors based on how participants define emotional variables (e.g., withdrawal, stress, craving, alcohol use) or environmental variables (e.g., location, the presence of other smokers) [5]. Furthermore, long-term self-reporting is more likely to be affected by the ‘Ostrich problem’ by which people avoid monitoring their behaviour, as it may be unpleasant, tiresome, or lead to unwanted changes in behaviour [9]. Therefore, collecting time information from mobile sensors can reduce the reliance on self-reports, and increase the accuracy of just-in time intervention messages [4].

Actions (including smoking) can be seen as being motivated by the need to maintain stability over time, in the face of a changing environment. This motivation can be interrupted by internal factors, e.g., feelings such as sadness, or external factors such as nicotine level [10]. A closed-loop control model is a common instrumental technique that seeks to maintain stability. It employs a feedback principle, using the output data from the model (feedback signal) as an input to modify the model’s actions, and hence maintain stability [11]. However, modelling addictive behaviour as a closed loop control model is a challenging task. It requires understanding the complexity of humans, as well as determining what elements should be counted to model the addictive behaviour. Moreover, when modelling the addictive behaviour, the goal state represents the fact that the system seeks to obtain a steady state (natural state), rather than to imply that there exists a single fixed value, as is often the case in system engineering [12,13].

Opponent process theory is claimed to be an essential method that can be used to model a person’s emotional state [14]. Solomon [15] described addictive behaviour using the opponent process theory. Within this model, an addict experiences pleasure as soon as a drug is supplied, which is followed by slowly accumulated withdrawal symptoms. As such, during the initial stages of addiction, the pleasure level is high and is accompanied by a low level of withdrawal symptoms. However, as time goes by, the withdrawal symptoms increase leading to a decrease in pleasure caused by using the drug, potentially resulting in a higher quantity of the drug being consumed [12].

Bobashev et al. [16] modelled the behaviour of smokers and employed the opponent process scheme of control theory. The model did not present any complex neurobiological process, only providing a mathematical model with a cascading feedback loop, aimed at presenting the scientific narrative of the opponent process as shown in Figure 1.

The model equations were developed with phenomenological interpretation in mind, and no real biological process was modelled. A set of continuous functions were used, feeding into the cascading functions. The system equations involve five interlinked processes,
(1)$$ProcessA:\frac{dY_{1}}{dt}=e^{-\alpha t}-b_{1}Y_{1}$$
(2)$$ProcessB:\frac{dY_{2}}{dt}=a_{1}Y_{1}-b_{2}Y_{2}$$
(3)$$ProcessC:\frac{dY_{3}}{dt}=a_{2}Y_{2}-b_{3}Y_{3}$$
(4)$$ProcessD:\frac{dY_{4}}{dt}=a_{3}Y_{3}-b_{4}Y_{4}$$
(5)$$ProcessE:\frac{dY_{5}}{dt}=a_{4}Y_{4}-b_{5}Y_{5}$$
where a, b and α are scaling coefficients, and all the $Y_{i}$ initial values are set to zero. Each equation presents a weighted integration of the previous one, causing the processes to lengthen successively. $Y_{1}$ represents the effect of nicotine level and is modelled with a pharmacokinetic equation. $Y_{2}$ represents the toxicity level and how the body processes the drug. $Y_{3}$ is the daily smoking habit. $Y_{5}$ is a longer scaling habit, which is scaled in years (rather than minutes/hours/days). While the process $Y_{4}$ has not been interpreted, it has been used to add scaling period between $Y_{3}$ and $Y_{5}$, which results in a slow change in process $Y_{5}$. To simulate smoking behaviour, a threshold value was defined to prompt self-administration. The threshold
(6)$$T=\frac{\left(\beta_{3}Y_{3}+\beta_{5}Y_{5}\right)}{\left(1+\beta_{2}Y2\right)}$$
has calibration coefficients $\beta_{i}$, and to avoid division by zero one is added to the denominator of the equation. The threshold value is changed based on external stressors to initiate cigarette use
(7)T=T+stress.

The research also modelled the withdrawal and craving processes; these processes begin immediately following the initial nicotine use and grow over time
(8)$$W=\frac{d_{3}Y_{3}\left(T-Y_{1}\right)}{\left(Y_{0w}+Y_{1}\right)}$$
(9)$$C=\frac{d_{5}Y_{5}\left(T-Y_{1}\right)}{\left(Y_{0c}+Y_{1}\right)}$$
where $d_{3},d_{5},Y_{0w}$ and $Y_{0c}$ are calibration coefficients. This control theory model was able to simulate plausible changes in smoking behaviour over time. However, the system was not able to present real-life behaviour, and could not capture individual differences between smokers’ daily habits. Figure 2 shows an example of the differences between the smoking behaviour as presented using the simulated control theory model Figure 2a and real-life data collected from a participant shown in Figure 2b.

Studies show that modelling smoking behaviour is essential, as it can improve the intervention process in the way of helping smokers in their most needed time [17]. While control theory models lack in prediction but provide an explanation, on the other hand, the deep learning (DL) models provide superior prediction without explanation. In order to get better time-series data prediction, it is useful to incorporate a mechanical structure into a phenomenological statistical model [18]. Following this hypothesis, this research proposes a deep-learning model, which when combined with a control theory model of smoking, will be able to adapt to the smoker’s unique behaviour and predict future smoking events. The Bobashev et al. [16] model was chosen due to its ability to capture the nicotine effect using the pharmacokinetic equation. The model can be later employed to develop a smart mobile app that will send automated interventions. Here, we describe the implementation of this control theory model of smoking that is expanded to incorporate other factors affecting smokers’ smoking behaviour (e.g., geolocation and motion).

## 2. Classification Method: The 1D Convolutional Neural Network

In recent years, deep learning as sub-field of machine learning (ML) has attracted great interest from the scientific community. DL refers to a deep neural network that consists of a massive web of interconnected nodes (whose depth is more than a single hidden layer). The nodes are able to perform complex, non-linear, computation on a set of input features, and give a suggested solution as an output. This new structure has been used to resolve many complex computer science problems such as image and speech recognition, with better accuracy compared to previous approaches of ML [19,20]. Convolutional neural networks (CNN) are a type of feed-forward neural network, which dates back to the 1980s. CNNs are composed of a convolution operation followed by a pooling operation [21,22]. With the increase interest in DL, CNNs have been reintroduced and used in many applications [23]. The main advantage of a CNN is its ability to be applied on parallel methods, and its high ability to learn, ensuring that all stages of the computation are appropriate for the data and for each other. To solve a problem using CNN, one should try experimenting with different variables including the number of layers, kernel size, choice of an activation function, etc. [24]. 1D-CNN performs a convolutional operation on the local region of the input data using different kernels for the individual features. Also, the size of the local region can vary for different features (this is not possible with a 2D CNN). The 1D convolutional operation in layer *l*,
(10)$$y_{j,i}^{l}=f\left(\sum_{m}\sum_{n}X\left(j-m,n\right)K\left(m,n,i\right)+b\right)$$
where K is the multi-dimensional convolutional kernel, *i* is the kernel index, *b* is the bias and X and *y* are the input and output respectively, performs dot-products across the input [25]. In most models, a Deep CNN will use a rectified linear unit (ReLU) f(x)=max(x,0), instead of a traditional neural network (hyperbolic tangent, logistic sigmoid) activation function. ReLU is more efficient, simpler and allows non upper-bounded output values. Also, in order to improve the performance of the CNN, regularisation techniques may be used, which reduces the generalization error while preserving the training accuracy [26].

## 3. Data Collection and Processing

A mobile application was developed, that can collect signals from mobile sensors (e.g., movement and environment), as well as participants’ self-report of smoking events. Five smokers (all taking at least 5 cigarettes per day) were recruited, and were asked to report their smoking events for two weeks. In the pre-processing stage of the data, samples for each day were unified to 1440 sample per day (one sample per minute). To do so, three types of events were registered in the dataset: smoking, not-smoking and app-off (representing gaps in the dataset due to, for example, participant’s mobile phone being off). Figure 3a shows the frequency of events for each of the five participants. It is clear from the data that the classes are unbalanced, as there are far fewer smoking compared to non-smoking events. Overall, of the 1440 data samples per day less than 15 per day are smoking events, while the rest are either not smoking or app-off events. To overcome this limitation, the time periods for labelling was changed to include a 30-min window followed the smoking event rather than a 1-min window, hence reducing the ratio of smoking to non-smoking events. Furthermore, it is assumed in the model that app-off is a non-smoking event, to remain cautious. Figure 3b shows the frequency of events for each of the five participants after applying these changes.

The reported smoking events were then used as input to the control theory model of smoking, in order to calculate the nicotine levels and threshold value during the 13 day period (one 24 h period was dropped because it was made of two half-days, one at the start and the other at the end of the data collection period). Calculated data (e.g., nicotine level) along with collected data (e.g., light, GPS Location, activity labels etc.) were combined to form the dataset for each participant. The reported smoking events were the labels for the data set. Figure 4 illustrates the process of data collection.

### 3.1. Mobile App

Data collection took place using a mobile application developed for Android mobile users, using Android Studio (IDE). The main focus of the User Interface (UI) was to develop a user-friendly interface that provides no feedback to users, as so to avoid influencing their behaviour [27]. The UI was used to label smoking events, relying on participants’ self-reporting. Users could report smoking events either by pressing a button on the main layout of the app, or by pressing a Widget on the home screen of the smartphone as can be seen in Figure 5.

The application was designed to run as a background service, which records data from the phone’s sensors. This service was designed to restart itself whenever terminated (either by the OS or otherwise). This was implemented in order to overcome a new restriction forced by Android on the development of background services that run for long periods. Collected data, along with smoking events were stored on an internal SQLite database.

### 3.2. Data Collection

For this study, the participants were healthy smoking adultsover 18 years old, with a good level of English literacy. They each owned and regularly use an Android mobile phone. Smokers were defined as those smoking at least 5 cigarettes a day for at least 6 months; they all smoke traditional cigarettes. During the data collection period, the application was installed on the participant’s smartphone for two weeks. No restrictions were been placed on their daily activities, and they were only asked to report their smoking events and keep the GPS on. At this stage of the research data has been collected from 5 participants (3 females: 2 male); all from the UK. The exclusion criteria were being under 18 years or over 55 years; self-reported physical or mental health issues that impact movement; not using an Android phone (e.g., using an iPhone). Although the number of participants appears small, The study by Schick et al. [6] modelled smoking behaviour using 4 participants, hence 5 participants were a sufficient number to model smoking behaviour. In addition, the ML model is trained for each participant separately, where a large volume of data was collected from each participant (approximately more than 1000 smoking events and 18720 samples each participant), making it sufficient for modelling a machine learning problem.

Data were collected from several sensors in order to identify correlations between smoking events and the sensors reading. Table 1 shows the types of collected data. The goal to use the collected data to find the association between smoking events and environmental data, in order to inform the implementation of a machine learning model that can automatically predict smoking events based on the occurrence of internal and external predictors. Following data collection, it emerged that not all sensors are available in all mobile models. Therefore the plan was modified to use only the common sensors that appear in most of the mobiles, i.e., the accelerometer and GPS values.

## 4. Approach to Model Development

To design a machine learning model for smoking behaviour, the control theory model of smoking was combined with the 1D-CNN. Initially, each part of the model was analysed separately before reaching the final model.

### 4.1. Control Theory Model of Smoking

While the actual nicotine level cannot be measured without lab oratory testing, that requirement does not accord with the aim of the research (creating a model that can be employed in a smart smoking cession app). The output from the control theory model of smoking is accepted as a description of the behaviour of the endogenous stressors, where the nicotine level is increased with every cigarette taken, then decreases gradually over time untill the next smoking event.

Using the reported smoking events, nicotine concentration was calculated using the control theory model of smoking [16] as shown in Figure 6. Each peak in the figure represents a smoking event, followed by a gradual decrease in the nicotine level until the next smoking event.

Figure 7 shows the threshold values calculated using the control theory model. The peaks represent non-smoking periods, and the threshold value decreases as the number of cigarettes take per day increases. The control theory model also describes withdrawal and craving symptoms, as demonstrated over a 10-day period in Figure 8.

### 4.2. Classification of Smoker Behavioural Data

Two types of events occur in the collected dataset, which are labelled as smoking (1) and not smoking (0). In order to further verify the effectiveness of the system, three types of ML models were explored; Support Vector Machine (SVM), Decision tree (DT), and 1D-CNN. The three classifiers were implemented and tested using Python. The Scikit-learn library was used to implement SVM and DT, while 1D-CNN was implemented using the Keras neural-network library with Tensorflow-GPU in the background.

The classification methods were tested to see whether the classifier could detect the smoking events based on each smoker’s motion and location factors. The data was trained and tested for each participant individually using an iterative process, where one day was held for testing while the remaining 12 were used for training. This routine was repeated separately for each of the 13 days. Six features (3 raw accelerometer values: x, y and z, and three GPS values: longitude, latitude, and altitude) were used as input to the ML models. First, the classifiers were tested using only 3 accelerometer values Table 2. Then the classifiers were tested using only the GPS values Table 3.

Table 4 shows the classification accuracy based on using motion and location factors which are 3 accelerometer (x, y and z) values and GPS (longitude, latitude, and altitude).

It can be seen from the tables that in general the performance of the 1D-CNN is consistently more accurate than the other two classifiers. The results indicate that in order to model the smoker’s behaviour, the model has to be trained based on the individual behaviour for each person; and that both motion and location are important for predicting smoking events.

The 1D-CNN is implemented such that a sequence of 10 past observations is mapped as input to the model. Each input feature is passed as a 1D input to a separate model in parallel to the others, and at the end the output from these models is combined to get the classification output. The model architecture consists of 6 parallel models, each of 1D-convolutional layer with 64 filters, followed by one max-pooling layer and a flattening layer. These models combine to one dense layer with 30 nodes, the output of which is passed to a sigmoid activation function to produce the final output (see Figure 9).

Figure 10 shows the confusion matrix for each participant; it can be seen that the continuity of the data is an important factor that affects the prediction of smoking events. Despite this, the overall classification accuracy remains high (mean: 0.87, standard deviation: 0.080). The final model predicts nicotine levels that are much closer to the original nicotine levels for all participants, as described in the next section.

## 5. Results

After testing the three classification methods, the 1D-CNN was selected as the most suitable classifier to predict smoking events. The classifier predicts either smoking or non-smoking states, with the app off event being treated as non-smoking events. The point of the prediction was to see if the model can accurately forecast the nicotine level (rather than use the originally calculated values) using the combined control theory and 1D-CNN model, and then predict smoking events based on this predicted nicotine value. As the nicotine level is considered a changed value over time. The model uses the comparison between the nicotine level and the threshold value as the first indicator for the need to have a cigarette. The model then makes use of external stressors (accelerometer and GPS) as input to the DL model in order to make the final decision regarding the likelihood of a smoking event. The importance of using the 1D-CNN model as part of the control theory model is to ensure the capture of the endogenous factors which affect the smoker’s behaviour as presented by nicotine level inside the smoker’s body. This approach should ensure that no intervention messages are sent before the nicotine level as derived from Equation (Equation 1) decreases to a level that is below the threshold as derived from Equation (Equation 6).

The resultant combined model of DL and control theory is shown in Figure 11. Six features were used as input to the DL model (three raw accelerometer values: x, y and z and three GPS values: longitude, latitude, and altitude). Since all these values and their combinations are personalised for each participant, training needs to take place for each participant. Testing the data iteratively enabled us to compare the prediction level for different days of the week. Since the output of the model is forecasting the nicotine effect value over time, Mean Square Error is used as the error criterion to measure the performance of the model. This evaluation matrix has been previously used to evaluate time-series data [28]. The results of the Mean Square Error (MSE), Root Mean Square Error (RMSE), and Normalized Root Mean Square Error (NRMSE), represent the accuracy of predicted nicotine levels during week days (Table 5) and weekends (Table 6). In general, the model has the same performance throughout the week.

Figure 12 shows the predicted nicotine level from a randomly selected day for two participants. All 6 predictors were used as input to the system. The nicotine level was predicted during the closed-loop process; no pre-calculated data was used.

Although some smoking events were missed, the model in general reliably predicts the smoking behaviour of each of the participants. While accuracy of prediction of nicotine level is negatively affected by missed samples in the data set overall accuracy remains high. Figure 13 shows the predicted smoking events for a randomly selected day with a high level of missed samples.

In some cases the participant was cooperative in reporting smoking events in all days accept for one day. The model predicted several smoking events for that day, and we cannot be sure whether these are unreported smoking events or false alarms (Figure 14).

Overall, the model can predict smoking events with 0.2 accuracy in a 15 min window, 0.3 accuracy in a 30 min window, 0.5 in a 1-h window, and 0.8 in a 2-h window. Figure 15 displaysthe ROC curve for all dataset for 15 min, 30 min, 1-h window, and 2-h window. As far as we know, there are relatively few studies which explore the possibility of using machine learning to classify the factors that lead to smoking events, and all the previous research [7,8] rely on self-reporting and surveys, which make it hard to compare with our research since they use different experiment settings and different inputs. Even though this research has accomplished a better overall accuracy equals to 86.6% without relying on self-reporting of predictors.

## 6. Conclusions

In conclusion, DL was successfully applied to model smokers’ behaviour. The model design combines 1D-CNN with a control theory model of smoking, and the results are promising. Two predictors were used as input to 1D-CNN (motion and geolocation) to predict smoking events. This design was able to adapt to the behaviour of individual smokers, with an average of 87% overall accuracy. This is a preliminary model, with potentials of improvement in the future. It is expected that the accuracy of the system in predicting smoking events will be increased by improving the DL model by adding layers, and by taking advantage of other information such as the indoor smoking ban in the UK. It may also be possible to construct a combined model of individuals’ behaviour, using additional external data such as the addresses of the smoker’s work and home, as well as public information on the location of businesses such as bars and tobacco shops, which are likely to be associated with smoking. These additions to the model are currently under consideration. This model is aimed to be used by smoking cessation app; the app can be integrated in the future into smoking cessation treatments. While all previous work in this area, even if they used ML models, they continue to rely on self-reporting of the predictors. Using DL opens the door to the possibility of automatic predicting the smoker’s behaviour, and that in turn allows sending automated innovations based on the individuals’ behaviour.

## Figures and Tables

**Figure 1 sensors-20-01099-f001:**
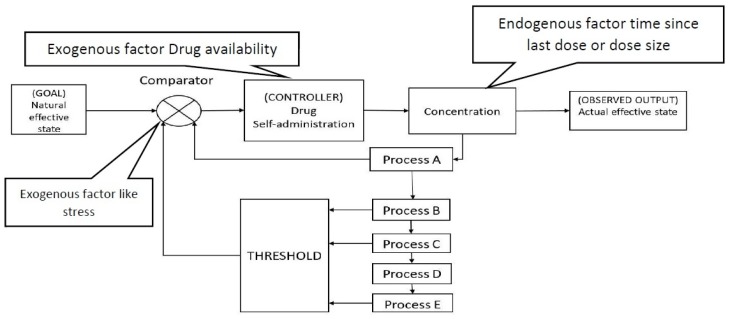
Control theory model of smoking re-drawn following [16].

**Figure 2 sensors-20-01099-f002:**
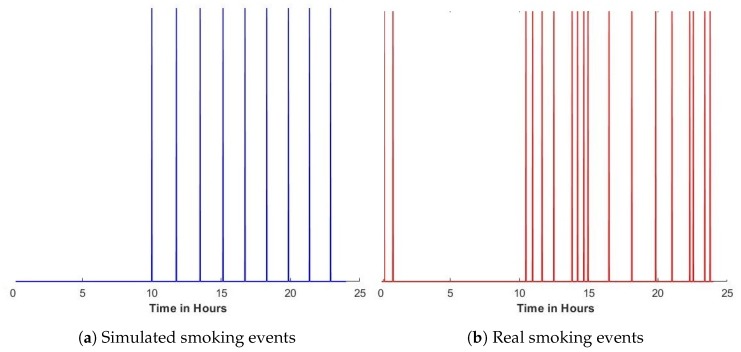
Smoking frequency; each peak represents a smoking event using smoking events reported from a randomly-selected day from our collected data of one of our participants. [2] (**a**) a simulated smoking behaviour generated by the control theory model [16], and (**b**) real smoking behaviour.

**Figure 3 sensors-20-01099-f003:**
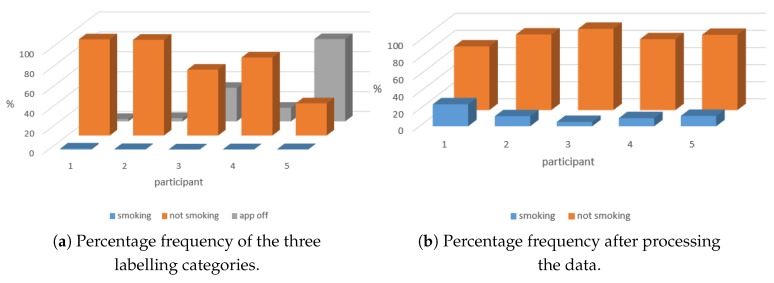
Data set Preprocessing.

**Figure 4 sensors-20-01099-f004:**
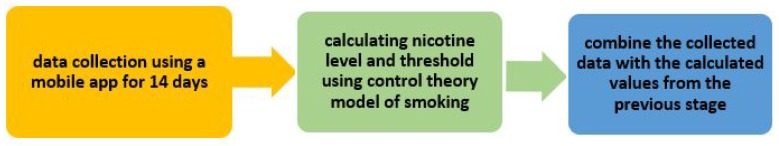
Overview of the study: data collection and processing steps.

**Figure 5 sensors-20-01099-f005:**
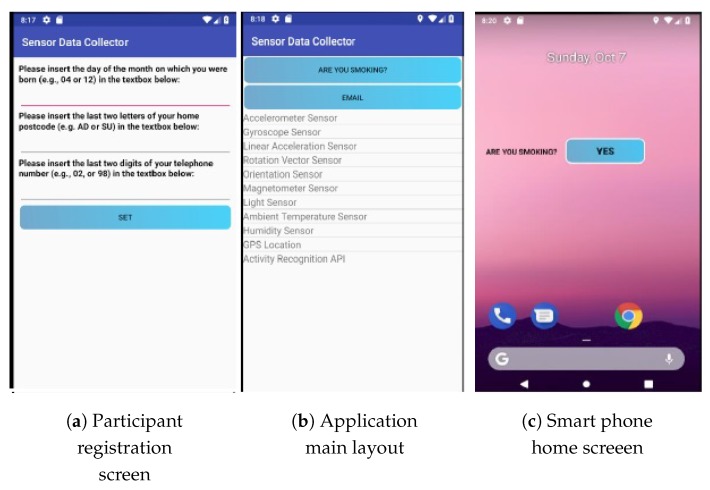
Mobile application User Interface (UI) [2].

**Figure 6 sensors-20-01099-f006:**
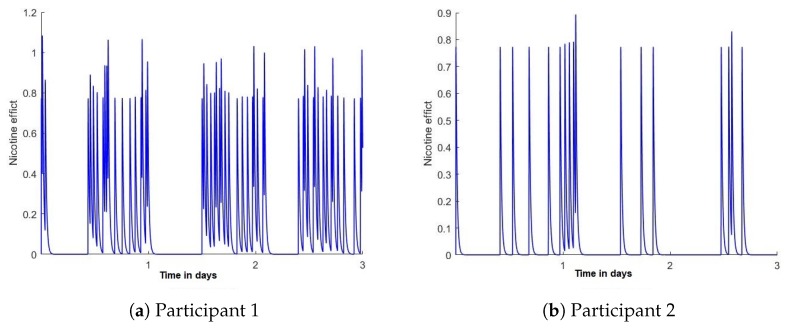
Examples of 3 days of smoking behaviour by two randomly selected participants, as modelled using control theory to represent nicotine levels [2].

**Figure 7 sensors-20-01099-f007:**
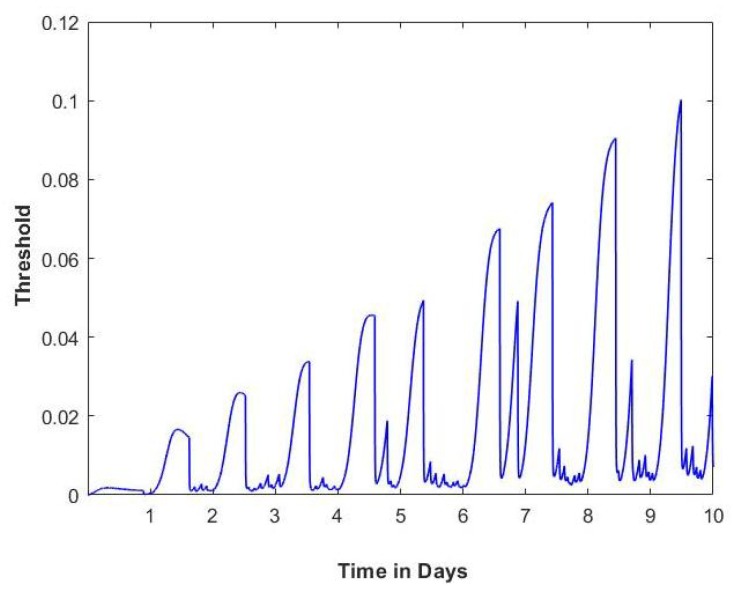
Example of 10 days calculated threshold value using the control theory model of smoking and collected data from one of the participants [2].

**Figure 8 sensors-20-01099-f008:**
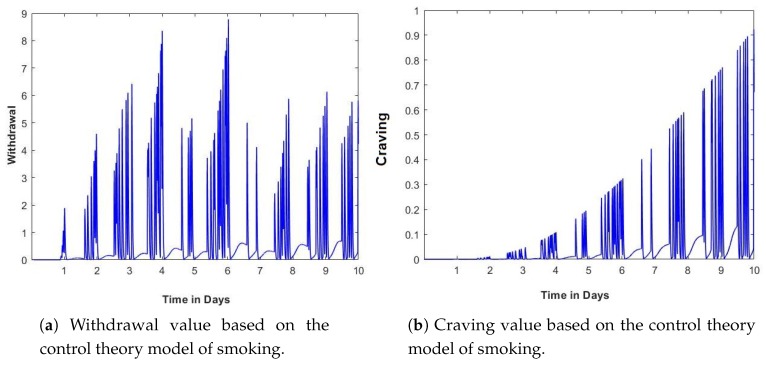
Example of 10 days calculated withdrawal and craving values using the control theory model of smoking and collected data from one of the participants [2].

**Figure 9 sensors-20-01099-f009:**
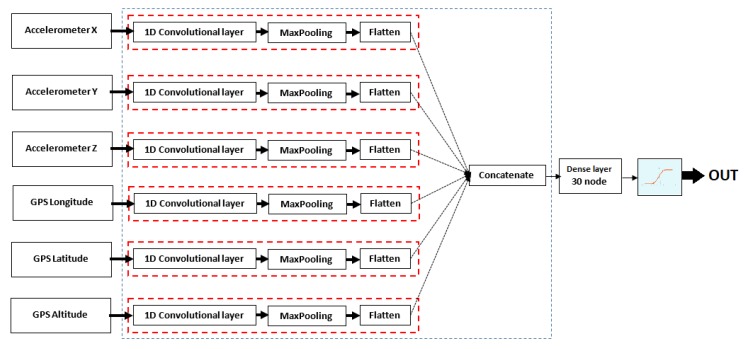
One-dimensional convolutional neural network (1D-CNN) model architecture.

**Figure 10 sensors-20-01099-f010:**
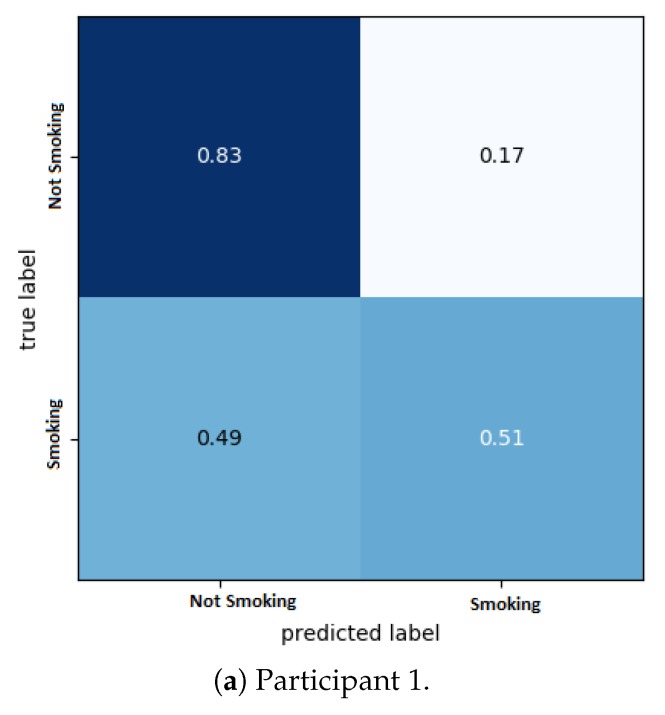
1D-CNN model confusion matrix.

**Figure 11 sensors-20-01099-f011:**
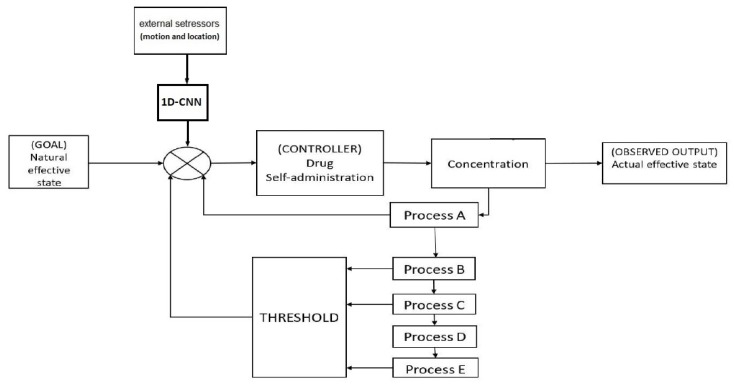
Smoking behaviour model utilizing machine learning. Data are collected and processed using the steps described in Figure 4. The 2 predictors are used as input to the 1D-CNN model. A classification value of 1 represents a potential smoking event. This value is passed to the CONTROLLER, simulating the taking of a cigarette, and re-initializing the parameters of the control model to zero.

**Figure 12 sensors-20-01099-f012:**
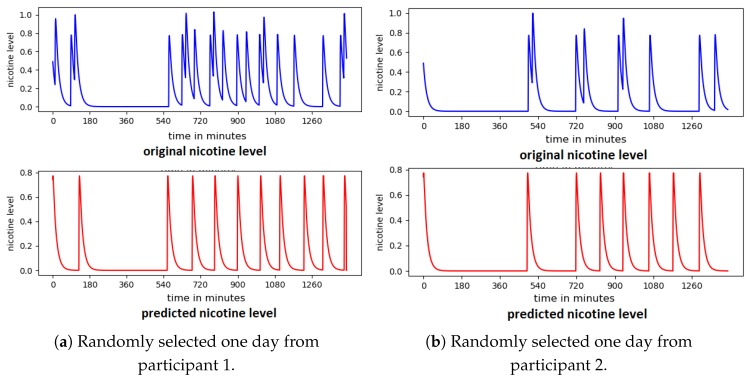
Predicted nicotine level from two participants.

**Figure 13 sensors-20-01099-f013:**
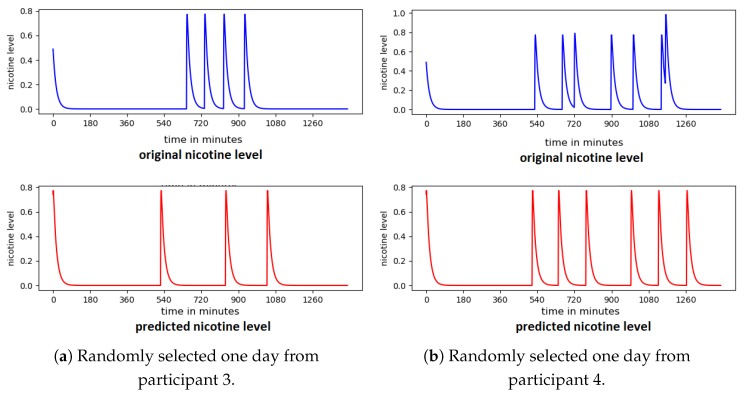
Predicted nicotine level for participants with a high presence of app-off values.

**Figure 14 sensors-20-01099-f014:**
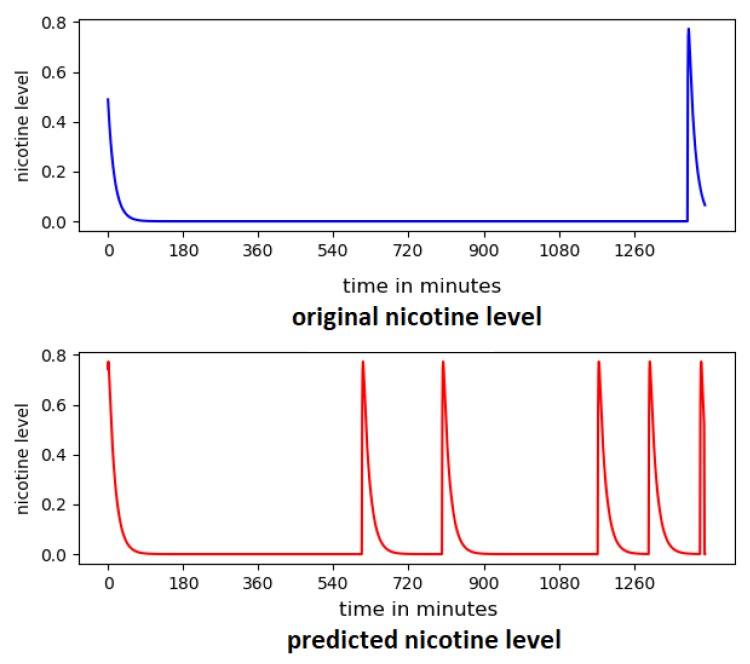
Predicted nicotine level for one day from participant 3 with no reported smoking events.

**Figure 15 sensors-20-01099-f015:**
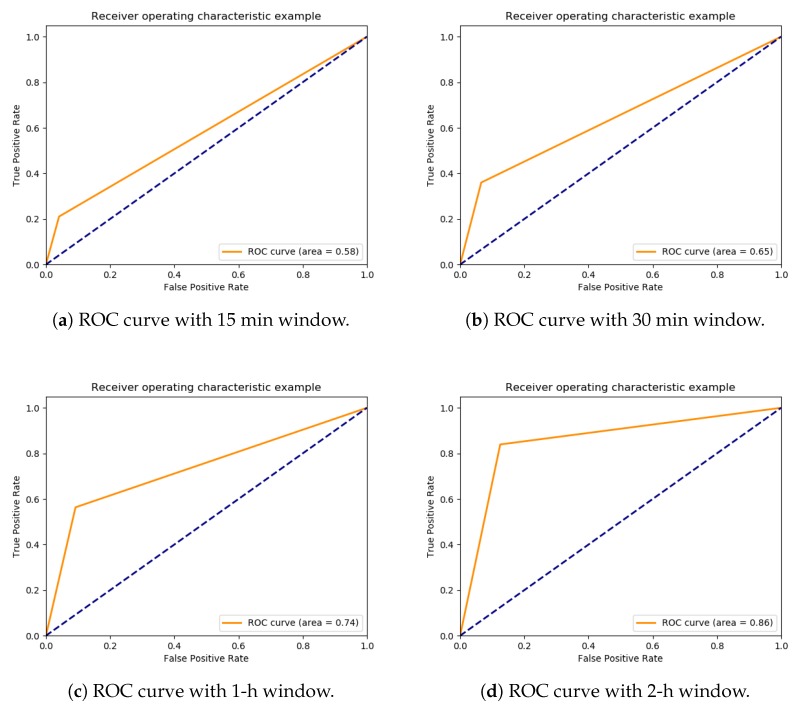
The deep learning model of smokers behaviour ROC curve.

**Table 1 sensors-20-01099-t001:** The number of labels in each of the three labelling categories.

Collected Data Group Name	Description
ID	This is unique ID that Identify the user data, it is set by the user at the start of the study.
Timing value	This is time stamp DD-MMYYYY, HH:MM:SS
Motion sensors data	Accelerometer, Gyroscope, Linear acceleration, Orientation, Rotation vector.
Environmental data	Magnetic field, Light level, Ambient temperature, Relative humidity, GPS location.
Activity labels	Google activity recognition API (Still, Running, Walking, Cycling, Tilting, and Driving).
Smoking labels	This is labelled by the user.

**Table 2 sensors-20-01099-t002:** The average accuracy based on only 3 GPS values.

Calculated Accuracy Category	SVM	DT	1D-CNN
Participant 1 smoking	0.01	0.40	0.01
Participant 1 not smoking	0.98	0.70	1.00
Participant 1 overall	0.73	0.62	0.74
Participant 2 smoking	0.03	0.51	0.09
Participant 2 not smoking	0.99	0.95	0.98
Participant 2 overall	0.88	0.90	0.87
Participant 3 smoking	0.02	0.08	0.00
Participant 3 not smoking	0.99	0.91	1.00
Participant 3 overall	0.94	0.95	0.95
Participant 4 smoking	0.00	0.24	0.08
Participant 4 not smoking	1.00	0.81	1.00
Participant 4 overall	0.90	0.88	0.90
Participant 5 smoking	0.00	0.25	0.21
Participant 5 not smoking	1.00	0.97	0.97
Participant 5 overall	0.88	0.97	0.88

**Table 3 sensors-20-01099-t003:** The average accuracy based on only 3 accelerometer values.

Calculated Accuracy Category	SVM	DT	1D-CNN
Participant 1 smoking	0.26	0.43	0.51
Participant 1 not smoking	0.74	0.75	0.83
Participant 1 overall	0.62	0.67	0.75
Participant 2 smoking	0.19	0.37	0.63
Participant 2 not smoking	0.88	0.89	0.95
Participant 2 overall	0.80	0.83	0.91
Participant 3 smoking	0.06	0.08	0.01
Participant 3 not smoking	0.95	0.94	1.00
Participant 3 overall	0.90	0.89	0.95
Participant 4 smoking	0.21	0.16	0.18
Participant 4 not smoking	0.91	0.93	0.97
Participant 4 overall	0.84	0.85	0.89
Participant 5 smoking	0.12	0.25	0.44
Participant 5 not smoking	0.95	0.95	0.94
Participant 5 overall	0.85	0.86	0.87

**Table 4 sensors-20-01099-t004:** The average accuracy based on all 6 featuers accelerometer and GPS values.

Calculated Accuracy Category	SVM	DT	1D-CNN
Participant 1 smoking	0.24	0.41	0.59
Participant 1 not smoking	0.79	0.77	0.79
Participant 1 overall	0.65	0.68	0.73
Participant 2 smoking	0.04	0.50	0.64
Participant 2 not smoking	0.87	0.92	0.94
Participant 2 overall	0.78	0.87	0.89
Participant 3 smoking	0.05	0.11	0.03
Participant 3 not smoking	0.96	0.93	0.99
Participant 3 overall	0.91	0.89	0.94
Participant 4 smoking	0.14	0.28	0.20
Participant 4 not smoking	0.91	0.87	0.97
Participant 4 overall	0.83	0.81	0.89
Participant 5 smoking	0.12	0.26	0.47
Participant 5 not smoking	0.95	0.95	0.94
Participant 5 overall	0.85	0.86	0.88

**Table 5 sensors-20-01099-t005:** The overall error rate of the proposed model during week days days.

	MSE	RMSE	NRMSE
Participant 1 smoking	0.07	0.26	0.23
Participant 2 smoking	0.05	0.22	0.20
Participant 3 smoking	0.02	0.14	0.15
Participant 4 smoking	0.03	0.18	0.21
Participant 5 smoking	0.03	0.15	0.17

**Table 6 sensors-20-01099-t006:** The overall error rate of the proposed model over the weekends.

	MSE	RMSE	NRMSE
Participant 1 smoking	0.07	0.27	0.24
Participant 2 smoking	0.03	0.16	0.17
Participant 3 smoking	0.01	0.10	0.11
Participant 4 smoking	0.03	0.16	0.20
Participant 5 smoking	0.01	0.12	0.15

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
