# Peer review of "Towards a Smart Smoking Cessation App: A 1D-CNN Model Predicting Smoking Events"

_sensors, 2020, doi:10.3390/s20041099_

Round 1
Reviewer 1 Report
This is a well-prepared manuscript presenting findings from the study on a model that can learn the smoker’s behavior. The study is novel and the findings are new and relevant for smoking cessation research. However, I have some concerns about the study design - participants.
The Authors did not provide sufficient “Characteristics of the study population”. There was information that five smokers (at least 5 cigarettes per day) were enrolled in this study. However, more details are needed, including (1) inclusion/exclusion criteria; (2) subjects characteristics (age, smoking history, smoking prevalence and other data related to the tobacco use); (3) nicotine dependence levels (do the authors assessed smoking dependence level e.g. with FTND?); (4) how the smoking status was assessed (self-reported or verified with appropriate measures?).
How about the number of participants? An explanation of the sample size (only 5 participants) will be helpful.
Moreover, in the Reviewer's opinion, the Discussion section is needed. The results of this study should be discussed concerning other studies in this field. Without the discussion section, it is difficult to assess the relevance of this paper. Some sentences on the potential implications of this paper and the possibility to apply the results into the clinical practice will be helpful.
Moreover, I have some minor comments:
1) Please consider providing more extensive and comprehensive abstract
2) Figure 2 - please clarify whether this is an original figure or it is based on another study? The data source for this figure is unclear.
3) Point 2 “1D Convolutional Neural network as Classification method” is a little bit confusing. It is a part of the background or methods section? Please separate it using appropriate headings.
4) Please provide potential clinical/practical implications of this paper
Reviewer 2 Report
A combination of mechanistic models with machine learning has been discussed for a while but not much have been done in this area and this development is definitely welcomed. The paper presents novel research that is very important and timely because of the high demand for explainable AI. Mechanistic models lack in prediction but provide explanation, while "black box" models provide superior prediction without explanation. So the combination of both is a very welcomed development.
Below are a few suggestions for revisions. They are mostly asking for clarification of methods and whether alternative methods were considered.
-Not clear why to use CNN, especially when only one layer is used. it is just a single layer neural network.
-Why use a CNN and not a recurrent neural network? RNN can take chunks of usage data and consider them as "patterns" to predict new outcomes/patterns?
-Was any time series analysis considered with the waiting time for the next smoke as the outcome? This might be a more natural way to use the mechanistic model, i.e. first to calibrate the model to the steady state corresponding to a particular subject and then to use both: the model-predicted time to the next event and CNN-predicted time to the next event, with CNN considering not just the current state but also the past events.
Bobashev, G. V., Ellner, S., Nychka, D. W., & Grenfell, B. (2000). Reconstruction of susceptible and recruitment dynamics from measles epidemic data. Mathematical Population Studies, 8(1), 1–29. doi: 10.1080/08898480009525471
-Why use MSE-based criterion as opposed to other options?
-The use of accelerometer data is probably covering the hours of sleep, but it might be useful just to add sleep hours explicitly as external “seasonality”. Similar things were done, for example, in
Ellner, S., Bailey, B., Bobashev, G. V., Gallant, A. R., Grenfell, B., & Nychka, D. W. (1998). Noise and nonlinearity in epidemics: Combining statistical and mechanistic modeling to characterize and forecast population dynamics. American Naturalist, 151(5), 425–440. https://doi.org/10.1086/286130
-Not clear how the mechanistic model was calibrated to the data. It is likely that different individuals will have different parameter settings. From the craving curve it looks like the control model was totally reset at time zero and not in a steady state. A steady state would be a natural state to assume for a regular smoker and this steady state will control craving from not getting out of control.
-Some editing for grammar and spelling will be helpful.
Round 2
Reviewer 1 Report
The Authors addressed all the comments that were previously mentioned. The methodology was improved.